# Life Cycle Assessment of Plywood Manufacturing Process in China

**DOI:** 10.3390/ijerph16112037

**Published:** 2019-06-08

**Authors:** Liangliang Jia, Jie Chu, Li Ma, Xuemin Qi, Anuj Kumar

**Affiliations:** 1College of Forestry, Northwest A&F University, Yangling 712100, Shaanxi, China; 18710803691@163.com (L.J.); mali111@nwsuaf.edu.cn (L.M.); 18392422980@sina.cn (X.Q.); 2Natural Resources Institute Finland (Luke), Production Systems, Tietotie 2, FI-02150 Espoo, Finland; anuj.kumar@luke.fi

**Keywords:** circular economy, Life cycle assessment, environmental hotspots, impact categories, sustainable strategy

## Abstract

Life cycle assessment (LCA) has been an important issue in the development of a circular economy. LCA is used to identify environmental impacts and hotspots associated with plywood manufacturing. Based on our results and a literature review of LCA studies involving plywood, a sustainable and environmentally friendly scenario was proposed for the plywood processing industry to improve environmental performance and sustainability. This study covers the life cycle of plywood production from a cradle-to-gate perspective, including raw material preparation and plywood manufacturing and processing to analysis of environment impacts and hotspots. Analysis of abiotic depletion (ADP), acidification effect (AP), primary energy depletion (PED), freshwater eutrophication (EP), global warming potential (GWP), and particulate matter (RI) were selected as major impact categories in this study. All data were obtained from on-site measurements (plywood production) and investigations of the Eco-invent database and CLCD database (upstream data of materials and energy). These data can be ignored when environmental contributions comprise less than 0.001% of environmental impact and auxiliary material quality is less than 0.01% of total raw material consumption. An eco-design strategy with eco-alternatives was proposed: pyrolysis bio-oil can be used to produce green resin to replace traditional phenolic formaldehyde (PF) resin to decrease the impacts of GWP, PED, AP, PM, and especially ADP and EP. A new technology of gluing green wood was used to replace conventional plywood production technology; wood waste could undergo a gasification process to produce resultant gas rather than combusting. Plywood was also compared with other wood-based panels in China to identify additional scenarios to improve environmental sustainability.

## 1. Introduction

Alongside the rapid development of the wood-based panel industry, China is one of the largest producers, traders, and consumers in the world. Jiangsu province is the second largest wood-based panel production region in China [1]. Various studies have presented life cycle assessment (LCA) of medium-density fiberboard (MDF), high-density fiberboard, particleboard or medium-density particleboard (MDP), oriented strand boards, and veneer-based products including plywood and laminated veneer lumber [2]. Plywood is a wood-based building product commonly used in China for commercial and residential construction. The plywood manufacturing process includes debarking, peeling, drying, sorting, gluing, laying up (compositing), and hot pressing. In 2017, plywood production in China reached 2.098 million m^2^, and its demand is increasing gradually every year [3].

LCA is an internationally accepted and standardized tool that considers resources consumed and emissions released along with corresponding environmental impacts. LCA is also used to assess environmental effects associated with a product [4,5]. LCA has been shown to be a valuable methodology in environmental management and preventative environmental protection; this method should be part of the decision-making process regarding sustainability. The applications of LCA pervade industrial society, such as in the design of industrial products and production processes, environmental policy making, and waste management as a parameter for cleaner production [6,7,8,9]. LCA has become the first choice for many countries when formulating industrial development strategies.

The main aim of the present study is to evaluate the environmental impact of traditional plywood manufacturing and determine environmental hotspots [10]. Based on the LCA method and eBalance software, we identified impact categories including abiotic depletion (ADP), acidification potential (AP), primary energy depletion (PED), freshwater eutrophication (EP), global warming potential (GWP), and particulate matter (RI) to analyze environmental hotspots. The LCA results can reveal areas of interest related to environmental impact, sustainability, and environmental scenarios. Findings can also inform recommendations to replace traditional plywood processing technology and materials in the plywood processing industry.

## 2. Goal and Scope 

### 2.1. Objective

From the preparation of raw materials and plywood processing, this study assessed plywood manufacturing from an LCA perspective to identify the environmental burden and provide parameters for cleaner production. We also offer data support and a research foundation for the establishment and development of a plywood LCA database. In this study, ADP, AP, PED, EP, GWP, and RI are analyzed based on the LCA method as impact categories to identify hotspots. 

### 2.2. Functional Unit

This paper used material from medium-sized plywood manufacturers in Suqian, Jiangsu Province, with an annual output of more than 600,000 m^3^ and fast-growing poplar as the wood species. The appropriate function unit was defined as 1 m^3^ of product. The conventional size of the board was 1220 × 2440 mm^2^, and its approximate moisture content was 7%.

### 2.3. System Boundary

This study covered the manufacturing life cycle of plywood from a cradle-to-gate perspective and ignored background data for raw materials (internal) and data on products sales (external). The product system is detailed in Figure 1 and includes the following main stages: raw material and energy production, transport, plywood production, and waste disposal. Boundary setting facilitated material tracking and cross-boundary energy flows. The system boundaries in this study included what occurred during production on-site along with off-site measurements, such as resources consumed in energy production, raw material production, additives and transport, and electricity generation; these features were partially collected and analyzed.

### 2.4. Data Quality

Table 1 lists upstream data to produce plywood, extracted from the Eco-invent database and CLCD database, and used to evaluate the cradle-to-gate life cycle of plywood in China.

All data in this study related to the input and outputs of plywood processing were obtained via on-site measurements and investigation of the Eco-invent and CLCD databases. Data on the consumption of materials, electricity, ancillary materials, water, and diesel came from on-site measurements used in the database. Key data included the technology used for plywood production and energy input and output per procedure.

Building a logical data processing method is also important for LCA assessment. Two data selection criteria were applied in this study—when the contribution of single-material input and per subsystem exceeds 0.001% of environmental impact, it cannot be neglected; and when the auxiliary material quality is less than 0.01% of total raw material consumption, it can be ignored.

## 3. Results of Impact Assessment

An analysis of LCA for plywood production was carried out according to the CML 2 baseline 2000 V2.4 method to quantify environmental impacts of the manufacturing process [11]. The LCA software eBalance (IKE, Chengdu, China) was used for environmental impact assessment. The total amount of life cycle inventory analysis (LCI) for each impact category can be calculated using the following formula (1):(1)$$LCI_{i}=\underset{p}{\sum }S_{p}\times \in V_{ip}\left(1\right)$$

In the above formula, *I* denotes the list of substances in the life cycle of the product (e.g., wood raw materials and water consumption); *LCI_i_*, denotes the total quantity of in the product life cycle; *P*, denotes a unit process in the product life cycle; *inV_ip_* denotes the number of list *I* in a unit process *P*; and *S_p_* denotes the process *P* of the baseline stream by the given LCA.

Several impact categories were analyzed in this study: ADP, AP, PED, EP, GWP, and RI. Results of the characterization step are displayed in Table 2.

To conduct an accurate LCA evaluation of plywood, we separated the processing of plywood production and preparation of raw materials to analyze environmental impacts as shown in Figure 2 and Figure 3. 

### 3.1. Abiotic Depletion Potential (ADP)

As Figure 3 indicates, among all materials, veneer production was responsible for ADP (86.53%). Among all plywood production processes, the subsystem of veneer compositing accounted for the highest ADP contribution (82.36%); veneer composting includes resin making, gluing, and aging. ADP is mainly caused by use of fossil fuels in the veneer-producing stage and use of phenolic formaldehyde (PF) in the composting stage.

### 3.2. Acidification Potential (AP)

In plywood processing, drying and compositing were the most important contributors to AC at 34.18% and 29.75%, respectively, followed by the plywood packing subsystem at 13.29% (Figure 2). This result is largely due to NO_X_ and SO_2_ emissions from the generation of heat energy and electricity consumption. Veneer production contributed 73% to the AC in preparation of all raw materials, mainly due to wastewater, combustion of waste wood, and fossil fuel.

### 3.3. Primary Energy Depletion (PED)

The debarking subsystem had the largest contribution to PED at 66.82%. Compositing and drying were responsible for 15.27% and 9.63%, respectively. The PED of the system was primarily caused by bark, representing remaining waste and other material waste as wood defects, such as burrows, dead knots, and slipknots. Veneer production contributed 98.76% to PED, mainly due to the consumption of fossil fuel and electricity.

### 3.4. Freshwater Eutrophication (EP)

PF resin was the main contributor to EP impact, causing the compositing subsystem to contribute most to this impact category at 60.55%; this was followed by board drying (15.26%) and debarking (11.13%). Veneer production was the main contributing sector to EP, with 78.23% of the impact of all materials production on the environment (as shown in Figure 3), followed by curing agents at 13.12%. This finding is mainly due to NO_X_ emissions during energy production.

### 3.5. Global Warming Potential (GWP)

The drying and compositing subsystems were responsible for GWP contributions of 33.18% and 33.68%, respectively. Veneer production was responsible for 76.41% of the impact category. CO_2_ was emitted from the combustion of waste wood and fossil fuel for generating heat energy, chemicals, and electricity.

### 3.6. Particulate Matter (RI)

The drying subsystem accounted for the largest contribution to the environmental profile at 38.96%. Veneer production was responsible for 82.25%, followed by curing agent production (10.20%). Detailed analysis of the RI considers emissions of PM2.5, NO_X_, and SO_2_. Two features warrant attention—the contributions of fossil fuel and coal consumption in processes such as chemicals, electricity, and thermal energy; and small amounts of RI were released from cutting and packing. Log debarking, log bucking, and sawdust handling were additional sources of RI emissions.

## 4. Discussion 

The results of impact assessment indicated that veneer production and the subsystems of drying and composting appeared to be the greatest contributors to environmental impact. Then, we conducted a detailed analysis of these environmental hotspots to consider alternate scenarios.

### 4.1. Environmental Hotspot Analysis

In the preparation of all raw materials, veneer production presented the highest contribution to all environmental categories (Figure 3). Veneer manufacturing consists of five main processes: Log debarking, log cut-off, softening of logs, peeling the logs into veneers, and drying the veneers [12]. The first step of debarking by debarking machines represents remaining waste in the forest industry; thus, the debarking subsystem was a primary contributor to all impact categories. Then, blocks were heated to approximately 93 °C (200 °F) using various methods (e.g., hot water baths, steam heat, and hot water spray with heat energy from diesel combustion). The log-softening stage was another important step in the emissions of CO_2_, CO, NO_x_, formaldehyde, and volatile organic compounds, especially in contributing to wastewater. When heating was completed, logs were processed to generate veneers by a veneer lathe. Finally, veneers were taken from the clipper to a veneer dryer, where they were dried to the required moisture content (between 6% and 15%). The veneer dryer acted as the main emission pollution source because heat from fossil fuel and by-product (waste material) combustion emits large quantities of organic compounds (e.g., CO_2_, CO, SO_2_, NO_x_, and formaldehyde) that greatly influence GWP, PED, AP, and RI. The stages of log debarking and bucking release filterable particulate matter of less than 10 micrometers in aerodynamic diameter (PM-10) along with organic compounds (AP) from steaming and drying operations.

The compositing subsystem made the greatest contribution (more than 40%) to all impact categories, followed by the drying subsystem at 22.4% (Figure 2). Composition of plywood panels proceeds as follows: Glue-making, gluing, assembly, and aging. The main sources of emissions include the gluing and aging processes, which release excessive amounts of formaldehyde and other hazardous air pollutants. 

The hot-pressing subsystem was responsible for a small share of impact categories (AP, PED, EP, GWP, and RI), mainly due to the release of organic compounds, benzene, formaldehyde, CO_2_, and SO2 [13,14]. However, some studies on wood-based panel manufacturing in other regions revealed different results, namely a larger contribution to environmental impact compared to this study [15]. As with the transportation of materials and products, this difference is mainly due to uncertainty regarding the environmental impact of hot pressing, the extent of emissions from hot pressing (depending on moisture content and veneer species), the type and amount of adhesive, and aging time after gluing [16].

### 4.2. Perspectives on Improving Environmental Assessment

This section presents ways to reduce pollution in the plywood process based on analysis of alternate scenarios and previous studies [5,17,18]. In this study, the three targets for environmental improvement are adhesive components (i.e., adhesive based on bio-oil), veneer drying, and compositing. 

#### 4.2.1. Change to Conventional Resin Consumption

In this study, petrochemical resins (phenolic) were used as adhesives in plywood production. The preceding discussion suggests traditional resin as a hot spot for formaldehyde emissions. This synthetic resin greatly influences primary environmental impact categories for two reasons: First, the composition of PF is basically of fossil origin, especially phenol obtained from fossil resources due to environmental factors related to the use of fossil fuels; and second, free formaldehyde is released into the air during the subsystems of veneer compositing, gluing, and hot pressing [19,20,21]. Therefore, we propose solutions to improve the environmental performance of plywood based on reducing consumption of PF resin and replacing petroleum-based phenol with bio-oil produced by biomass fast pyrolysis; such pyrolysis contains many phenolic compounds, such as phenol, guaiacol, and 4-methyl guaiacol that can be condensed with formaldehyde [22]. Sensitivity analyses were conducted in five scenarios that simulate replaced percentages of PF resin in the plywood manufacturing process: S1—original scenario in this study; S2—replace PF resin with 30% bio-oil; S3—replace PF resin with 40% bio-oil; S4—replace PF resin with 50% bio-oil; and S5—replace PF resin with 60% bio-oil. It is found that replacing phenol with bio-oil can reach 60% under a catalyst loading of 1.25 g of NaOH; results showed that dry shear strength and wet shear strength were comparable between pure PF and bio-oil-PF for plywood. Additionally, plywood produced by bio-oil-PF emitted lower formaldehyde emissions than traditional plywood [23].

When PF was replaced by up to 60%, impacts on plywood production decreased by 4.47% for ADP, 4.11% for EP, 2.74% for RI, 2.68% for GWP, and 1.67% for AP as shown in Figure 4. Replacing PF resin can thus evoke environmental benefits, especially in terms of diminishing EP and ADP impacts. Previous research on wood-based panels has found that using green bonding agents instead of PF to produce HB elicited similar results as those in this study [24]. Therefore, renewable resources should be encouraged to replace fossil energy to decrease impacts on ADP, EP, RI, GWP, and AP.

#### 4.2.2. Change Technology in Gluing and Drying

The subsystems of compositing and drying exerted major impacts on all environmental categories due to large releases of formaldehyde, NO_X_, SO_2_, CO_2_, and volatile organic compounds (Figure 2), which come from waste materials and fossil oil combusted to generate heat to dry veneers before gluing. New technologies can be tested to identify impacts on the environment. Gluing green wood is a new technology that can be applied in plywood vacuum molding wherein veneer-composited wood in a green stage is glued without drying because it is vacuum-dried when pressure is applied to the press [25]. This method reduces the number of subsystems in manufacturing and avoids release of free formaldehyde and other noxious emissions during compositing and gluing. Figure 5 shows the results of LCA for plywood production that can reduce the drying stage.

Results show that using the new technology of gluing green wood can promote improvements in all impact categories, especially for GI, GWP, and AP, which were reduced by 38.96%, 34.18%, and 33.18%, respectively. This new technology seems promising in decreasing the environmental impacts of plywood. When reducing the drying stage, traditional internal recycling of waste material (e.g., barks and edges) will be influenced. These forms of waste may represent a gasification process to produce resultant gas that could be utilized to generate electricity and process heat [26]. Compared with traditional combustion treatment, gasification could eliminate approximately 90.0% of particulate material and 50.0% of NO_X_ without releasing carbon monoxide.

## 5. Conclusions 

This paper focuses on the environmental impact of plywood production in south China. Several impact categories (ADP, AP, PED, EP, GWP, and RI) were assessed to obtain results and serve as decision-making indicators to help the wood-based panel industry develop and introduce alternatives in plywood processing to improve the environmental performance of production. According to our results, in the process of plywood production, manufacturing of veneers in all raw materials had the greatest impact on the environment, mainly attributed to the drying stage of the veneer manufacturing process. The compositing stage was the largest contributor in all subsystems to all impact categories for the environment, followed by log debarking due to fossil fuel combustion as energy and bark waste. This study covers the plywood manufacturing process from a cradle-to-gate perspective and analyzes each subsystem and raw material in the plywood production process. To improve environmental performance, advanced technologies and green materials can be used instead of traditional processes. In this study, we suggested that pyrolysis bio-oil replace phenol (nonrenewable) to produce green phenolic resin to decrease contributions in impact categories including GWP, PED, AP, and PM during the plywood manufacturing process, especially to lessen the impacts of ADP and EP. In addition, the new technology of gluing green wood can reduce the veneer-drying stage and decrease NO_X_ and CO_X_ emissions generated by fossil fuel and waste wood (containing adhesive) combustion during veneer production. This environmentally friendly technology can also decrease all impact categories, especially RI, GWP, AP, and EP. Moreover, the location of the factory near a rich supply of quality wood resources can control environmental impact substantially.

## Figures and Tables

**Figure 1 ijerph-16-02037-f001:**
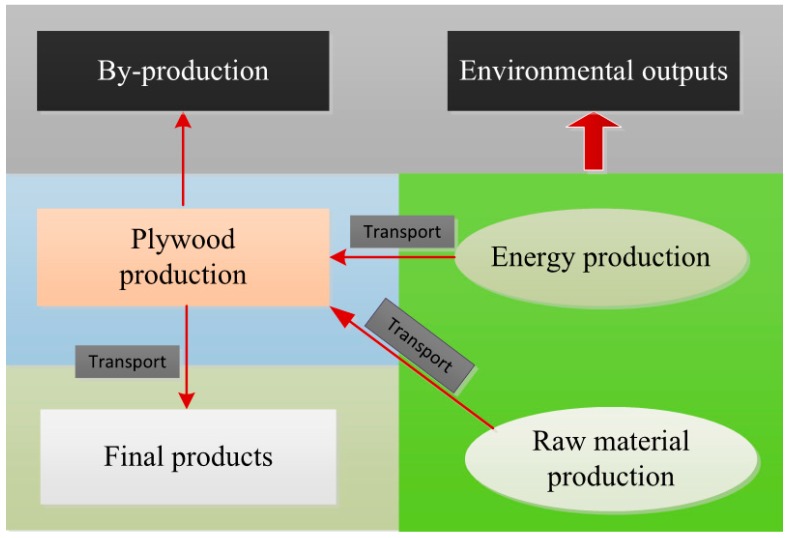
System boundary of plywood manufacturing process of life cycle assessment (LCA).

**Figure 2 ijerph-16-02037-f002:**
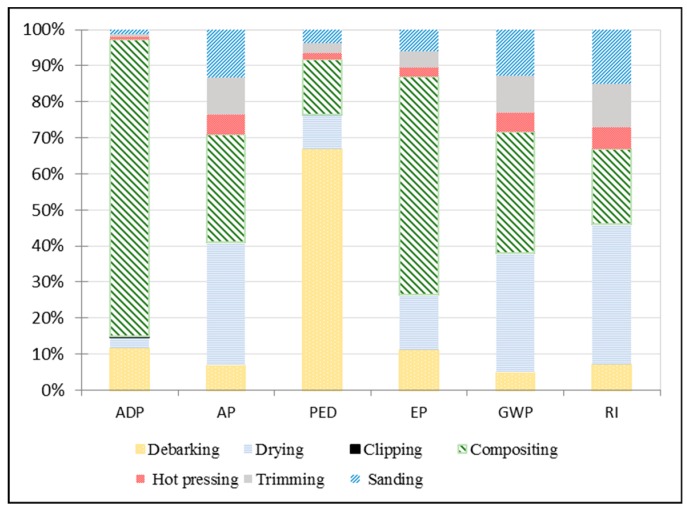
Contribution per subsystem (in %) to each impact category.

**Figure 3 ijerph-16-02037-f003:**
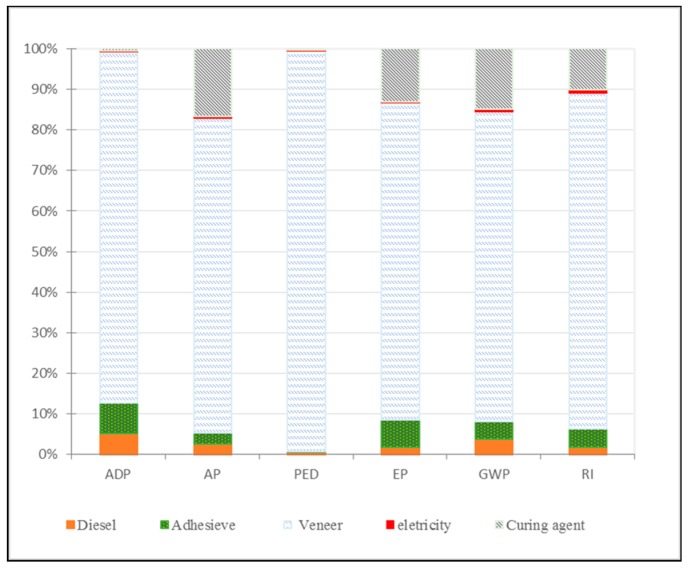
Relative contributions of material production (%) to each impact category; veneer includes debarking and drying stages.

**Figure 4 ijerph-16-02037-f004:**
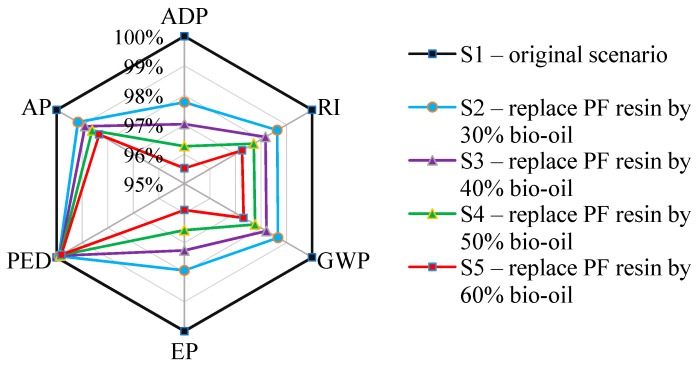
Sensitivity analysis for reducing consumption of phenolic formaldehyde (PF) resin.

**Figure 5 ijerph-16-02037-f005:**
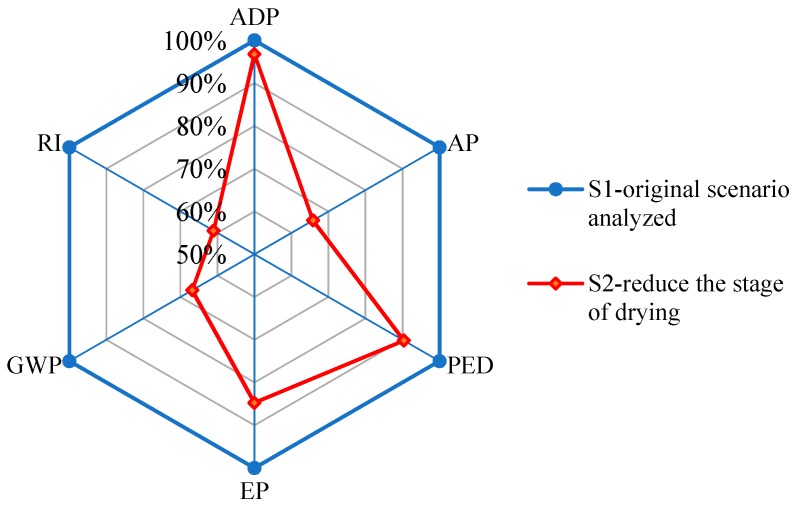
Analysis to reduce drying stage.

**Table 1 ijerph-16-02037-t001:** Upstream data of plywood products.

Name	Standard	Productive Place	Years	Database
Log	logs, mixed, at forest	Europe	2010	Eco-invent
Resin	melamine formaldehyde resin	Europe	2010	Eco-invent
Curing agent	synthetic ammonia	China	2013	CLCD
Veneer material	logs, mixed, at forest	Europe	2010	Eco-invent
Paint	resin size, at plant	Europe	2010	Eco-invent
Water	Industrial water	China	2013	CLCD
Electric	state grid	China	2013	CLCD
Steam	steam production (1 Mpa, 183 °C)	China	2013	CLCD
Diesel	market average	China	2013	CLCD
Transport	truck transportation(46t)-diesel	China	2013	CLCD

**Table 2 ijerph-16-02037-t002:** Impact assessment results (characterization step) of plywood manufacturing for 1 m^3^ of finished plywood.

Impact Category	Unit	Value
ADP	kg antimony _eq._ ^*^	2.90 × 10^−1^
EP	kg ${\mathrm{PO}}_{4}^{3-}$ _eq._	3.32 × 10
RI	kg PM2.5 _eq_.	3.43 × 10
AP	kg SO_2 eq_.	1.38 × 10^2^
GWP	Kg CO_2 eq_.	1.88 × 10^4^
PED	MJ	9.85 × 10^6^

* Indicator for assessing product and measure unit.

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
