# Peer review of "Life Cycle Assessment of Plywood Manufacturing Process in China"

_ijerph, 2019, doi:10.3390/ijerph16112037_

Reviewer 1 Report

  Life cycle assessment (LCA) is an internationally accepted and standardized process tool that evaluates resources consumed and emissions released along with the environmental impacts and widely used to assess the environmental facts for products. Nowadays, LCA has been also considered important methodology for evaluating sustainability in much fields.

   In this paper, the life cycle assessment to whole production process of plywood factory in southern China that evaluated and analyzed by the method of LCA, and both the data analysis and the environmental impact assessment of  plywood including  the  consumption of resource, energy and environmental indicators of one cubic meter plywood were obtained. It will provide methodological guidance for low carbon development of  the international wood industry in future. In general, the technical route of the paper is scientific and reasonable, the conclusions are accurate . So  I suggest that this paper will be revised and can be accepted.

  Suggestions and improvement as follows:

1. Please give a brief introduction to the relevant corporate production information of plywood in the article.

2.  Please describe the wood species (for example, softwood or hardwood species) of plywood to better reflect the background data of the life cycle evaluation.

3. It is noted that the manuscript has few mis-spellings and needs careful editing to format of words.

Author Response

Dear reviewer for my article,

Thanks for your honest and meaningful advice for my paper and I have studied it carefully and made some changes in it. The main ones are as follows:

Suggestions:
(1) Please give a brief introduction to the relevant corporate production information of plywood in the article

 response :The corporate production information of plywood has been given in line 69-72.

  (2)  Please describe the wood species

response : Wood species of plywood is fast-growing polar  that   describe in line 71.

  (3)  Editing to format of words

 The paper has been edited again by LetPub. And the author professor Kumar Anuj (Luke) has check it .

Reviewer 2 Report

In this manuscript, the authors analyzed the life cycle of plywood production and suggested that environmentally friendly technologies (such as new technology of gluing green wood) and green materials should be used in order to improve environmental performance. This paper may be accepted in International Journal of Environmental Research and Public Health with minor revision. The following points should be clarified:

1) The authors misused the full term and its abbreviation style, e.g. depletion potential (ADP), exhibited many times in the manuscript including Line 14, Line 53, Table 2, Line 109, Line 114. Please check all the other full terms, which is used once when it first appears in the manuscript.

2) In Line 5, and line 167, “Shanxi” should be changed into “Shaanxi”.

3) Please keep the numeric format consistent in Table 2. In the value column, two types of format- E+01 and 106 were used.

4) There are many typos in the manuscript. Please add space between these words.

For example:

In Line 40, “standardizedprocess” should be “standardized process”;

In Line 41, “toolthatconsiders” should be “tool that considers”;

In Line 45, “LCApervade” should be “LCA pervade”;

In Line 45, “(RI)to” should be “(RI) to”;

Please check all the words in the manuscript.

5) Please change the reference format according to the Reference List and Citations Guide of this journal.

Author Response

Dear reviewer for my article,

Thanks for your honest and meaningful advice for my paper and I have studied it carefully and made some changes in it . The main ones are as follows:

suggestions:
1) The authors misused the full term and its abbreviation style, e.g. depletion potential (ADP), exhibited many times in the manuscript including Line 14, Line 53, Table 2, Line 109, Line 114. Please check all the other full terms, which is used once when it first appears in the manuscript.

 response:

 The paper has been check in the whole part.

2) In Line 5, and line 167, “Shanxi” should be changed into “Shaanxi”.

 The paper has been check in the line 5 and line 398.

3) In the value column, two types of format- E+01 and 106 were used

  The paper has been used in  E+01.

4) check all the words in the manuscript.

 The paper has been edited by author Anuj.

5) Please change the reference format.

The paper has been edited according to this journal.

Reviewer 3 Report

This paper is interesting and contributes to LCA of wood products.However, when reading the paper, I often have wished to get more well written and concise sentences and sometimes I missed the stringency.

I would suggest the authors to revisit again this paper. It is interesting idea and topic. However, it is not well written. I would suggest them to ask English native speaker, English service or the authors should put more efforts to revise by themselves to improve for the sake of readibility.

The authors should revisit again the introduction, goal and scope, result, discusion and most importantly conclusion section. But overall, it is interesting idea and topic. Below you can find my comments.

Author Response

Dear reviewer for my article,

   Thanks for your honest and meaningful advice for my paper  and I have studied it carefully and made some changes in it. The main ones are as follows:

suggestions

(1)never use the word “potential” anymore because we already multiplying them with our inventory results.

Response: We have eidt it in line 14.

(2)Introduction 

Line 45. You used old reference in 2009.

Response: It has been replaced by new reference in 2019 in line 74.

how to distinguish between impact and potential (GWP, EP, ADP and AP)

Response:

Life Cycle Assessment mainly assesses potential environmental impacts,for example,potential accounts for percent of environmental impact.  The drying and compositing subsystemswere responsible for GWP contributions of 33.18% and 33.68% respectively.(Line 282-283).

(3) Goal and Scope

 1)use proper way to explain impacts. 

Response: it is same above.

2)system boundary

Response: This study coveredthe manufacturing life cycle of the plywood from a cradle-to-gate perspective and ignored background data for raw materials (inside) and data ofthe products sale(outside).( line 98-99)

3)table2 

Response: it is same above.

4) LCA is just environmental assessment not sustainability assessment as a whole.

Response: It has been edit as  environmental assessment in line 354,417,418.

Conclusion

Line 253. The authors claim to cover the whole life cycle. It is not true because the system boundary is only cradle to gate not cradle to grave or cradle to cradle. So, it is just part of the life cycle, not whole.

Response: It has been edit that is same above.

 In addition,  the paper has been edit by  editorbar that is English native speaker 

Thank you .

Best regards,

     Jie chu

    5.31.2019